# Checkpoint kinase 1/2 inhibition potentiates anti-tumoral immune response and sensitizes gliomas to immune checkpoint blockade

Crismita Dmello [1,2,18,19] ✉, Junfei Zhao[3,4,18], Li Chen[1,2], Andrew Gould[1,2], Brandyn Castro[1,5], Victor A. Arrieta[1,2,6], Daniel Y. Zhang [1,2], Kwang-Soo Kim[1,2], Deepak Kanojia[1,2], Peng Zhang[1,2], Jason Miska [1,2], Ragini Yeeravalli[1,2], Karl Habashy[1,2], Ruth Saganty[1,2], Seong Jae Kang [1,2], Jawad Fares[1,2], Connor Liu[7,8], Gavin Dunn[7,8,9], Elizabeth Bartom [10], Matthew J. Schipma[11], Patrick D. Hsu[12,13,14], Mahmoud S. Alghamri[15,16], Maciej S. Lesniak[1,2], Amy B. Heimberger [1,2], Raul Rabadan [3,4,17], Catalina Lee-Chang [1,2,19] ✉ & Adam M. Sonabend [1,2,19] ✉

Whereas the contribution of tumor microenvironment to the profound immune suppression of glioblastoma (GBM) is clear, tumor-cell intrinsic mechanisms that regulate resistance to CD8 T cell mediated killing are less understood. Kinases are potentially druggable targets that drive tumor progression and might influence immune response. Here, we perform an in vivo CRISPR screen to identify glioma intrinsic kinases that contribute to evasion of tumor cells from CD8 T cell recognition. The screen reveals *checkpoint kinase 2* (*Chek2*) to be the most important kinase contributing to escape from CD8 T-cell recognition. Genetic depletion or pharmacological inhibition of Chek2 with blood-brain-barrier permeable drugs that are currently being evaluated in clinical trials, in combination with PD-1 or PD-L1 blockade, lead to survival benefit in multiple preclinical glioma models. Mechanistically, loss of Chek2 enhances antigen presentation, STING pathway activation and PD-L1 expression in mouse gliomas. Analysis of human GBMs demonstrates that Chek2 expression is inversely associated with antigen presentation and T-cell activation. Collectively, these results support Chek2 as a promising target for enhancement of response to immune checkpoint blockade therapy in GBM.

Gliomas and glioblastoma (GBM) in particular, are the most common malignant primary brain tumors in adults[1]. Unfortunately, despite extensive research and the use of multimodal therapeutic strategies, the median overall survival time is still relatively short[1]. These tumors have a complex relationship with the immune system and rely on multiple mechanisms of immune suppression to inhibit anti-tumor immune responses including the induction of T-cell anergy, exhaustion, apoptosis, and sequestration in the bone marrow[2–18]. Immune checkpoint inhibitors have shown remarkable responses in some malignancies[19], especially those that are heavily infiltrated with T cells

A full list of affiliations appears at the end of the paper. ✉e-mail: crismita.dmello@northwestern.edu; catalina.leechang@northwestern.edu; adam.sonabend@northwestern.edu

and have a high tumor-mutational burden. However, despite the use of a wide variety of immune therapeutic strategies such as vaccines, dendritic cells, adjuvants, adoptive cellular therapies, and immune checkpoint inhibitors, immunotherapy has not shown efficacy for gliomas[20].

In recent analyses of recurrent GBM patients that were treated with PD-1 immune checkpoint blockade, we showed that mitogen-activated protein kinase (MAPK) signaling activation in tumor cells was associated with response to this therapy[21,22] highlighting the role of tumor kinases in immune modulation. Kinases are often altered in cancer-driving intracellular signaling cascades[23]. Moreover, kinases are targetable with small molecule inhibitors, and some kinase inhibitors are blood-brain barrier (BBB) permeable. Thus targeting kinases is a potentially relevant translational approach to the modulation of anti-tumoral immunity for GBM[24]. A previous in vitro CRISPR screen identified the SWI/SNF chromatin remodeling complex in tumor cells as an epigenetic modulator of T-cell recognition through the IFN-γ pathway[25]. Using an in vivo CRISPR screen approach to investigate the causal relationship between kinases and anti-tumoral immunity is relevant since it can capture the influence of the tumor microenvironment.

In this work, to identify tumor intrinsic kinase that contributes to the evasion of CD8 T cells in an unbiased fashion, we perform an in vivo kinome knockout (KO) CRISPR screen evaluating 713 kinases in glioma cells intracranially implanted into wild-type (WT) and CD8 KO mice. KO of checkpoint kinase 2 (*Chek2*), a kinase whose canonical function relates to DNA-damage response, shows the highest depletion among all the kinases, in tumors implanted into immune-competent mice relative to CD8 KO hosts. In other words, out of all the kinases, knockout of Chek2 in glioma tumor cells sensitizes them to CD8 T-cell-mediated killing. In this study, we characterize the effect of Chek2 inhibition/depletion on the response to PD-1 or PD-L1 blockade in the murine glioma models. The immune responsiveness upon Chek2 inhibition/depletion shows association with enhanced antigen presentation and increased type I interferon response manifested by low *Chek2* expressing tumor cells in preclinical models and in human GBM scRNA-seq datasets.

## Results

### In vivo kinome knockout CRISPR screen identifies kinases implicated in CD8 T-cell immune evasion

CD8 T cells are the predominant effector population in cancer mediated immunity[26]. To interrogate the interaction between tumor intrinsic kinases and CD8 T cells in gliomas, we first performed immune profiling to characterize immune cell types in WT and CD8 deficient (CD8 KO) mice. Our flow cytometry data confirmed the absence of CD8 T cells in CD8 KO mice, whereas all other immune cell types such as CD4 T cells, NK cells and macrophages remained unchanged between the WT and CD8 KO mice (Supplementary Fig. 1a, b). Furthermore, there were no survival differences between WT and CD8 KO C57BL/6 mice implanted with GL261 glioma cells (n = 9/group, p = 0.2) (Fig. 1a) which remained consistent with the prior studies[27]. To investigate the contribution of glioma cell-intrinsic kinases in T-cell recognition, GL261 cells were intracranially implanted after they had been transfected with a KO CRISPR library for all 713 known kinases so that a single kinase was deleted per cell (Fig. 1b). To identify the relative contribution of each of these kinases in evading CD8 T-cell-mediated killing, the kinome KO glioma cells were implanted in both WT and CD8 KO C57BL/6 mice (Fig. 1b). As mice approached the endpoint, they were euthanized and the tumor region was excised, genomic DNA was extracted, and the guides were amplified and sequenced (Supplementary Fig. 2a, b). A difference in survival, trending towards significance, was seen between WT (n = 11) relative to CD8 KO mice (n = 9; p = 0.06), implanted with kinome KO GL261 glioma cells (Fig. 1c), suggestive of CD8 T-cell-mediated selection in WT mice implanted

with kinome KO GL261 glioma cells. The normalized kinase gRNA read counts were used to calculate fold change depleted and fold change enriched in WT relative to the CD8 KO mice group (Fig. 1d). Kinase KO clones that were enriched in WT mice as compared to the CD8 KO mice were the kinases that contributed towards susceptibility to CD8 T-cell cytotoxicity (Fig. 1e) while kinase KO clones depleted in WT mice relative to the CD8 KO mice contributed towards resistance to CD8 T-cell-mediated killing (Fig. 1f). Glioma clones with *Chek2* KO were the most depleted sgRNAs in mice with intact immunity as compared to the CD8 KO mice (p < 0.0001).

To further validate our initial CRISPR screen, we performed a second, independent CRISPR screen using GL261 glioma model where we quantified the Chek2 KO clones in intracranial gliomas in WT and CD8 KO mice hosts over time. We found negative selection of Chek2 KO clones in mice with WT CD8. As shown in Fig. 2a, b, we collected samples at early stage and late stage of the screen. Similar to the first CRISPR screen, the second CRISR screen demonstrated that the selection of Chek2 KO glioma cells over time is specific to WT mice and is absent in CD8 KO mice (Fig. 2c–e). In the first CRISPR screen, 3/4 *Atm* (an upstream activator of Chek2) sgRNAs had normalized counts <5, which prevented further analysis (Fig. 2f). However, CRISPR screen 2 showed similar depletion of *Atm* sgRNAs over time as Chek2 (Fig. 2g, h) in WT mice group as compared to the CD8 KO group. However, unlike Chek2, selection of its functionally related kinase Chek1, was not specific to WT mice (Fig. 2i–k). In other words, Chek1 KO clones were significantly depleted in both WT and CD8 KO mice compared to the non-targeting sgRNAs (Fig. 2l, m). The second CRISPR screen validated our initial CRISPR screen findings.

### Tumor cell intrinsic *CHEK2* inversely correlates with type I interferon response in human GBMs and dampens this response in mouse glioma cells

To understand the phenotype of T cells and tumor cells associated with low and high *CHEK2* expression in tumor cells, we interrogated the single-cell RNA sequencing (scRNA-seq) data of 28 GBM patients[28]. We analyzed 7930 cells and stratified tumor cells as expressing high or low levels of *CHEK2* using the median as cutoff value. In this dataset, cells were categorized as tumor cells, macrophages, oligodendrocytes, or T cells based on the expression of specific gene sets (as described by Neftel et al.[28]) (Fig. 3a). *CHEK2* appeared to be primarily expressed in macrophages and tumor cells of the human GBM specimens (Fig. 3b, c). Next, we analyzed the gene expression pattern of T cells associated with low and high *CHEK2* expressing tumor cells. T cells from the low *CHEK2* expressing tumor cells showed gene ontology (GO) term enrichment for "Interferon γ (IFN-γ) signaling" as compared to the T cells from high *CHEK2* expressing tumor cells (p = 0.0005; Fig. 3d) (Supplementary Data 1). On the tumor-cell compartment, the low *CHEK2* expressing tumor cells showed GO term enrichment for "Interferon type I response" as compared to the tumor cells from high *CHEK2* expressing tumor cells (p = 0.032; Fig. 3e). Further, we analyzed another independent scRNA-seq dataset to assess the phenotype of the tumor cell in the context of *CHEK2* expression[29]. This scRNA-seq data also showed enrichment of "Interferon type I response pathway" (p = 2.8e−13) in low *CHEK2* expressing tumor cells as compared to the high *CHEK2* expressing tumor cells (Supplementary Fig. 3a, b). To investigate whether correlations seen in human GBM patients have a causal relationship, we generated a Chek2 KO clone in the GL261 mouse glioma cell line using the CRISPR cas9 system. A scrambled guide control was generated and designated as a non-targeting control (NTC). The KO was confirmed using western blotting (Fig. 3f). Next we treated Chek2 KO GL261 clones with IFN-γ to test its responsiveness to IFN-γ signaling. We found significantly increased surface expression of PD-L1, a known IFN-γ inducible gene[30], on the Chek2 KO relative to the control cells upon stimulation with IFN-γ (p = 0.003;

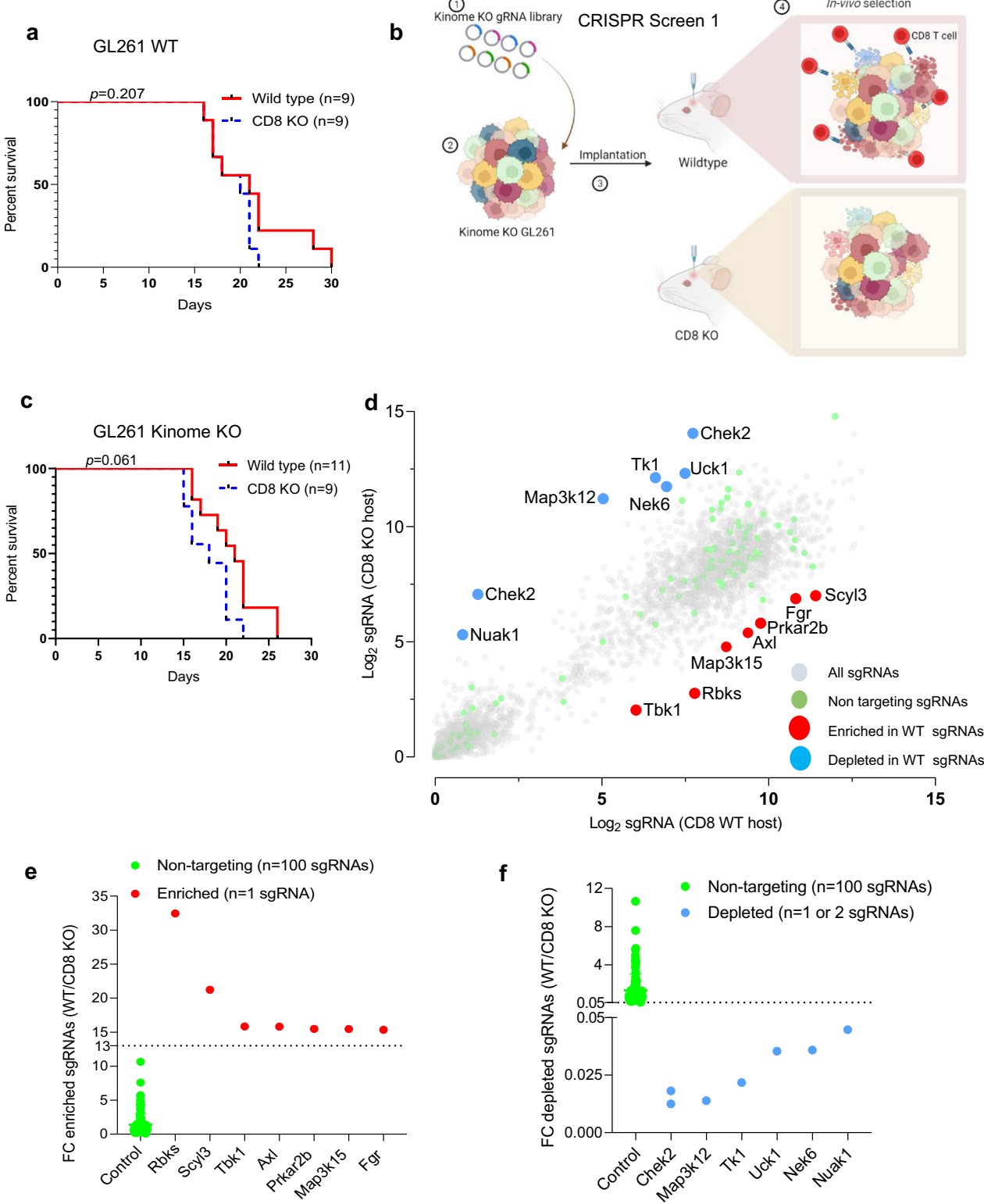

Fig. 3g) (Supplementary Fig. 4). Along the same lines, Chek2 KO clones demonstrated increased baseline mRNA expression levels of type I interferon-response genes IFN-β, IRF7, ISG15, except PD-L1 and IFNα as compared to the NTC. Moreover, following IFN-γ treatment, expression of type I interferon-response genes by quantitative real-time PCR analysis was further increased in the Chek2 KO as compared to the NTC cells (Fig. 3h). This data is suggestive of a negative correlation between Chek2 expression and response to IFN-γ signaling.

**Tumor cell intrinsic *CHEK2* inversely correlates with antigen presentation pathway in human GBMs and impairs this pathway in mouse glioma cells**

The scRNA-seq data of GBM patients[28] showed enrichment of antigen presentation pathway in low *CHEK2* expressing tumor cells as compared to the high *CHEK2* expressing tumor cells ($p = 3.2e−08$; Fig. 4a). Correspondingly, the T-cell compartment associated with the low *CHEK2* expressing tumor cells showed GO term enrichment for

**Fig. 1 | In vivo kinome knockout CRISPR screen in CD8 KO and WT mice. a** KM survival curves of the C57BL/6 WT and CD8 KO mice (*n* = 9/group) bearing GL261 glioma. The median survival durations in the groups were: WT, 21 days; CD8 KO, 20 days; Statistics: WT versus CD8 KO, log-rank test *p* = 0.2. **b** Schematic representation of the in vivo CRISPR screen. Mouse glioma cells GL261 were transduced with kinome knockout library and the transformed cells were implanted in WT and CD8 KO mice. The library representation of >500X was maintained in the WT and CD8 KO group by injecting $2 \times 10^5$ cells/mouse. As the animals approached the endpoint, they were sacrificed, and the genomic DNA was extracted from the tumor region, guides were amplified, and sequenced. **c** KM analysis of the animals in the CRISPR screen 1. The KM plot shows percent survival of WT (*n* = 11) and CD8 KO (*n* = 9) animals bearing kinome KO GL261 glioma cells. The median survival durations in the groups were as follows: WT, 21 days; CD8 KO, 18 days; Statistics: WT versus CD8 KO, log-rank test *p* = 0.06. **d** Scatter plot showing the top kinases with the most enriched or depleted sgRNAs in the WT as compared to the CD8 KO animals. The sky blue dots correspond to the top depleted sgRNAs, while the red dots represent the most enriched sgRNAs in the WT as compared to the CD8 KO animals. The gray dots are all other sgRNAs. **e** The *Y*-axis shows fold change (fc) WT over CD8 KO of the normalized sgRNA counts. The red dots correspond to the top enriched sgRNAs and the fluorescent dots are all non-targeting sgRNAs. **f** The Y-axis shows fc WT over CD8 KO of the normalized sgRNA counts. The blue dots correspond to the top depleted sgRNAs and the fluorescent dots are all non-targeting sgRNAs. For **e**, **f** the error bars on non-targeting sgRNAs represent mean ± SD. Source data for **a**, **c**–**f** are provided as a Source Data file.

"T-cell-mediated cytotoxicity pathway" as compared to the T cells from high *CHEK2* expressing tumor cells (*p* = 0.042; Fig. 4b) (Supplementary Data 1). Moreover, in the Abdelfattah et al.[29] scRNA-seq data we also observed an a gene signature enrichment of the antigen presentation pathway (*p* = 1e−11) in low *CHEK2* expressing tumor cells as compared to the high *CHEK2* expressing tumor cells (Supplementary Fig. 3c). Further, the T-cell compartment associated with the low *CHEK2* expressing tumor cells showed significant enrichment in pathway related to T-cell proliferation (*p* = 5.1e−17) as compared to the T cells from high *CHEK2* expressing tumor cells (Supplementary Fig. 3d). Thus, *CHEK2* expression in tumor cells was associated with consistent differences in the phenotype of T cells in GBM.

As in the case of *CHEK2*, glioma cell expression of *ATM*, the upstream activator of *CHEK2*, also exhibited a T-cell phenotype similar to *CHEK2* expression. The T-cell compartment associated with the low *ATM* expressing tumor cells showed GO term enrichment for "Interferon γ (IFN-γ) signaling" (*p* = 0.00019) and "T-cell-mediated cytotoxicity pathway" (*p* = 0.017) as compared to the T cells from high *ATM* expressing tumor cells (Supplementary Fig. 5a–c).

Unlike *CHEK2*, the T-cell compartment associated with the low *CHEK1* expressing tumor cells showed no significant enrichment for "Interferon γ (IFN-γ) signaling" (*p* = 0.76) and "T-cell-mediated cytotoxicity pathway" (*p* = 0.18) as compared to the T cells from high *CHEK1* expressing tumor cells (Supplementary Fig. 6a–c). Further, to investigate whether these correlations seen in human GBM patients have a causal relationship, we transfected the antigen reporter ovalbumin into both NTC and Chek2 KO cells in mouse glioma cell line GL261. As shown in Fig. 4c (Supplementary Fig. 7), we evaluated the ability of Chek2 depleted glioma cells to present MHCI restricted OVA peptide-SIINFEKL and activate SIINFEKL-specific OT-I CD8+ T cells. We found the enhanced presentation of MHCI-bound SIINFEKL peptide on the surface of IFN-γ treated Chek2 KO glioma cells as compared to NTC cells (Fig. 4d). Concomitantly, IFN-γ treated Chek2 KO glioma cells induced enhanced proliferation of OT-I CD8+ T cells as compared to the NTC cells ("OT-I CD8 T" panel, Fig. 4e). This phenomenon was predominantly dependent on antigen presentation, as IFN-γ treated Chek2 KO glioma cells failed to activate CD8+ T cells from WT C57BL/6 mice ("WT CD8 T" panel, Fig. 4e) (Supplementary Fig. 8).

Besides the immune-modulatory function of Chek2 demonstrated here, Chek2 is known to be involved in the DNA-damage response (DDR) and inhibition of Chek2 impairs DNA-repair pathways[31]. We investigated the changes in the induction of DNA-damage response using phosphorylation of γH2A.X as an indicator of DNA-damage response. Chek2 KO showed no difference in the phosphorylation of γH2A.X at the baseline as compared to the NTC cells (Supplementary Fig. 9a). Next, to investigate if Chek2 KO led to an increase in point mutations that could lead to the generation of more neoantigens, paired RNA and exome sequencing was performed for Chek2 KO and NTC cells (Supplementary Fig. 9b). Chek2 and Laptm4b were the only two genetic alterations predicted to have high binding affinities to MHC class I and gene expression (FPKM > 1) (Supplementary Fig. 9c, d). Identification of the Chek2 gene as one of the predicted

neoantigens was expected since CRISPR cas9 KO of Chek2 introduces a frameshift mutation leading to generation of truncated transcript that is detected as a neoantigen. Contribution of Laptm4b to Chek2-related immune response needs further investigation.

## Chek2 depletion/inhibition is associated with the STING pathway activation and upregulation of surface levels of PD-L1

Next, we investigated the mechanism/s underlying the enhanced gene expression of antigen presentation and type 1 interferon (IFN1) in glioma cells associated with Chek2 loss. We found that Chek2 inhibition/depletion resulted in activation of the STING pathway, as marked by the phosphorylation of TBK1 and upregulation of PD-L1 surface levels in GL261 and NPA glioma cell lines (Fig. 5a–g) (Supplementary Fig. 10). Further, we found a positive correlation between STING pathway and T-cell infiltration in the TCGA GBM dataset (Fig. 5h). These results suggested that Chek2 depletion/inhibition in tumor cells activates STING pathway and upregulates PD-L1 expression therefore, we hypothesized that Chek2 depletion/inhibition will sensitize gliomas to PD-1 or PD-L1 blockade, given that these forms of immunotherapy are thought to involve activity of effector T cells (Fig. 5i).

## Chek2 depletion sensitizes gliomas to PD-1 blockade immunotherapy

Given the immune evasive function of Chek2 in glioma cells, we investigated whether Chek2 expression could influence glioma response to PD-1 blockade. We first confirmed that the WT GL261 line implanted into immune-competent mice, did not respond to PD-1 blockade (Fig. 6a, b). Next, we performed a similar survival study with GL261 NTC and Chek2 KO glioma cells. No statistically significant improvement in survival was seen in mice implanted with GL261 NTC controls treated with anti-PD-1 as compared to IgG treated animals (*p* = ns, Fig. 6c). Significant improvement in survival was seen in mice injected with GL261 Chek2 KO gliomas treated with anti-PD-1 as compared to IgG treated animals with 30% long-term survivors (LTS) (*p* < 0.05, Fig. 6d). The LTS animals (*n* = 3) in the anti-PD-1 group that were rechallenged with the same cells, on the contralateral hemisphere, showed no tumor growth as compared to the control animals (Fig. 6e). Overall, genetic depletion of Chek2 in tumor cells showed a modest improvement in response to PD-1 blockade in mouse GL261 glioma model.

## Pharmacological inhibition of Chek1/2 sensitizes gliomas to PD-1 blockade

To explore the translational potential of Chek2 modulation in the context of anti-PD-1 immunotherapy in gliomas, the blood-brain barrier permeable Chek1/2 inhibitor Prexasertib (LY2606368) which is being evaluated in clinical trials[32–35], was tested in vivo. Compared to the vehicle control group, the combination of Prexasertib and anti-PD-1 conferred a significant extension of survival and 30% of animals were cured of glioma (Fig. 7a, b). Rechallenging of the LTS animals with tumor cells implanted in the contralateral hemisphere led to the rejection of the glioma, indicating induction

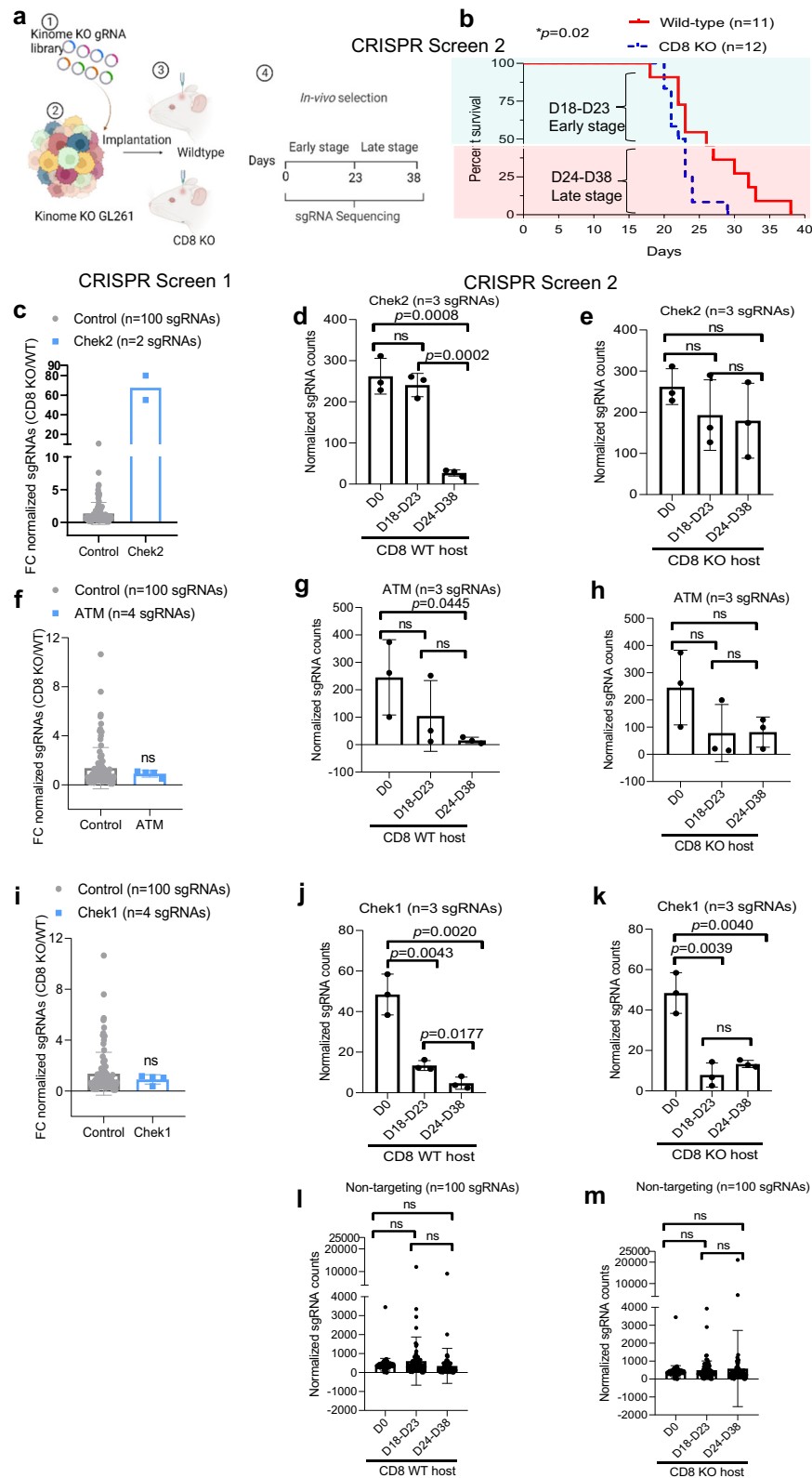

of immunological memory and surveillance (Fig. 7c). Immunophenotyping of the rechallenged LTS animals showed an increase in the percentage of CD8 T-cell population expressing pro-inflammatory cytokines IFN-γ and TNF-α as compared to control tumor-bearing mice, in the brain compartment (Fig. 7d). Furthermore, splenic CD8 T cells from the LTS animals, treated with the combination therapy, expressed significantly higher levels of both IFN-γ and TNF-α,

relative to non-tumor-bearing and control tumor-bearing mice (Fig. 7e). No such trend was seen in the CD8 T cells from deep cervical lymph nodes (dCLN) of the LTS animals treated with the combination therapy as compared to the non-tumor-bearing and control tumor-bearing mice (Fig. 7f). Similarly, we performed a survival experiment using ATM inhibition (AZD1390) in combination with PD-1 blockade. Survival of mice treated with combination

**Fig. 2 | In vivo kinome knockout CRISPR screen with early and late time points shows selection of Chek2 KO glioma cells over time in intact immunity mice as compared to CD8 KO mice. a** Schematic representation of the second in vivo CRISPR screen. The library representation of 733X was maintained in the WT (*n* = 11) and 800X in the CD8 KO (*n* = 12) group by injecting 200,000 cells/mouse. Different barcodes were assigned to the animals that were sacrificed at the early stage (D18-D23) and animals sacrificed at the late stage (D24–D38) from both WT and CD8 KO hosts, guides were amplified and sequenced. **b** KM analysis of the animals in the second CRISPR screen. The KM plot shows percent survival of wild-type and CD8 KO animals bearing kinome KO glioma cells. *p* = 0.02 using the log-rank test. **c** The histogram shows 74-fold enrichment of Chek2 sgRNAs in CD8 KO mice as compared to the WT mice in the CRISPR screen 1. In the CRISPR screen 2, **d, e** Change in Chek2 KO glioma cells over time in WT and in CD8 KO mice respectively. **f** The histogram shows change in Atm sgRNAs in CD8 KO mice as compared to the WT mice in the CRISPR screen 1. In the CRISPR screen 2, **g, h** Change in Atm KO glioma cells over time in WT and in CD8 KO mice respectively. **i** The histogram shows fold change in Chek1 sgRNAs in CD8 KO mice as compared to the WT mice in the CRISPR screen 1. In the CRISPR screen 2, **j, k** Depletion of Chek1 KO glioma cells over time in WT and in CD8 KO mice respectively. The distribution of non-targeting sgRNAs over time in WT mice (**l**) and CD8 KO mice (**m**) of the CRISPR screen 2. For **c**–**m** the error bars represents mean ± SD. For **d**–**m** statistics was done using unpaired two-tailed *t*-test (without adjustments for multiple comparisons). Source data for Fig. **b**–**m** are provided as a Source Data file.

of ATM inhibition and PD-1 blockade was similar to PD-1 blockade alone, but in contrast to the survival study using Prexasertib, no differences in the fraction of LTS was observed between these groups (Supplementary Fig. 11). Overall, our findings based on CRISPR screen, survival studies with PD-1 blockade, and scRNA-seq analysis suggest that the immune-modulatory phenotype of the ATM/Chek2 pathway is most robust when Chek2 is directly targeted as opposed to indirect, upstream targeting.

## Combination therapy of Chek1/2 inhibition and PD-1/PD-L1 blockade shows efficacy in the context of standard-of-care radiotherapy

Radiation is a standard of care for glioblastoma patients. The Chek1/2 inhibitor AZD7762 was selected for these experiments since it has shown efficacy in the context of radiation therapy[36]. We used genetically engineered model derived cells called as NPA-with genetic features (NRAS, shP53, shATRX, wt-IDH1)[36] that resembles human gliomas[37,38]. First, we tested the responsiveness of this model to PD-1 and PD-L1 blockade and found no significant improvement in survival (Supplementary Fig. 12a, b). The combination of AZD7762 + anti-PD-1 with standard-of-care radiotherapy and AZD7762 + anti-PD-L1 with standard-of-care radiotherapy showed significant extension of survival compared to the radiotherapy treated vehicle control group, with 30% of mice showing tumor eradication (Fig. 8a, b). Similar findings were observed in GL261 tumor-bearing mice showing a significant extension of survival as compared to the radiotherapy treated vehicle control group with 40% of mice showing tumor eradication in combination of radiation+AZD7762 + anti-PD-1 group and 60% of mice showing tumor eradication in combination of radiation+AZD7762 + anti-PD-L1 group (Fig. 9a, b). Radiotherapy alone was able to extend the survival of GL261 glioma-bearing mice by 10 days (Supplementary Fig. 13). To ascertain the role of CD8 T cells in mediating therapeutic response, the same treatment regimen was used in GL261 tumor-bearing CD8 KO mice. There was no significant difference in survival between the combination group of radiation+AZD7762 + anti-PD-1 as compared to the other groups, in CD8 KO mice (Fig. 9c), attributing the therapeutic effect seen in radiation+AZD7762 + anti-PD-1 combination therapy, to CD8 T cells.

## Discussion

We describe Chek2 kinase as an immune modulating kinase that contributes to tumor cell evasion from CD8 T-cell cytotoxicity in gliomas. Chek2 as a target, was identified in this study using an in vivo kinome CRISPR KO screen, was the most depleted kinase in WT immune-competent mice relative to CD8 KO mice. This suggested that tumor intrinsic Chek2 may be regulating resistance to CD8 T-cell-mediated cytotoxicity. Our mechanistic studies and human correlative analysis supports our CRISPR screen-derived observation that targeting this kinase can lead to activation of T cells. Our study describes the potential therapeutic value of combining a Checkpoint kinase 1/2 inhibitor with immune checkpoint blockade in murine glioma models. This is relevant since clinical trials investigating the efficacy of immune

checkpoint blockade in GBM have failed to show a survival benefit[21,39–42].

ScRNA-seq analysis of GBM samples in two independent datasets showed negative correlation between tumor intrinsic Chek2 expression and type I interferon response and antigen presentation on tumor cells. A negative association was observed between tumor intrinsic Chek2 expression and IFN-γ signaling and cytotoxicity of T cells. Along these lines, type I interferon response and antigen presentation have been shown to promote anti-tumor CD8 T-cell responses[43]. Evaluation of immune-modulatory effects of the Chek1/2 inhibitor Prexasertib demonstrated an increased expression of T-cell activation related genes and decreased expression of immunosuppression-related genes in mouse head and neck squamous cell carcinoma[44]. Blosser et al. identified interferon alpha and gamma response genes as gene sets associated with resistance to Prexasertib, using transcriptome analysis of a pan-cancer cell line panel, sarcoma and neuroblastoma xenograft models[45]. We found that Chek2 inhibition or depletion results in STING pathway activation and upregulation of PD-L1 expression which perhaps leads to increased expression of type I IFN genes that mediates susceptibility to killing by CD8 T cells.

Chek1 and Chek2 have been previously found to be overexpressed in recurrent GBMs post radiotherapy[46]. As such, Chek2 expression might contribute to the ineffectiveness of immune checkpoint blockade in GBM patients. The canonical function of Chek2 is to participate in DNA-damage responses[47] yet our Chek2 KO clones did not display any difference in DNA-damage response. However, our results suggest that the mechanism by which Chek2 inhibits anti-tumoral immunity maybe due to STING pathway inactivation and not due to modulation of DNA-damage response since we did not observe any difference in the frequency of neoantigen predictions or increase in somatic mutations upon depletion of Chek2. Although CRISPR screen targeting is associated with frameshift mutations and potential generation of neoantigens, it is unlikely that such possibility explains the immune-modulatory effects we observe with Chek2 loss, since 1. Both Chek2 KO and inhibition of Chek2 kinase activity (with pharmacological inhibitors) demonstrated the same effect on STING pathway activation and expression of PD-L1 in two different glioma cell lines. 2. IFN-γ treated Chek2 KO glioma cells induced enhanced proliferation of OT-I CD8+ T cells while it failed to activate CD8+ T cells from WT C57BL/6 mice. 3. At least 2/4 *Chek2* guides targeting different regions on the Chek2 gene were significantly depleted in WT mice as compared to the CD8 KO mice in 2 independent CRISPR screens. This is unlikely to occur if the selection is due to neoantigen generation given the random DNA insertions and deletions associated with CRISPR KO.

In our study, pharmacological inhibition of Chek1/2 by Prexasertib and AZD7762, two different inhibitors proven safe in clinical trials, rendered gliomas susceptible to PD-1 blockade in glioma models. As such, combinatorial treatment strategies of Chek1/2 inhibitors with immune checkpoint inhibitors and radiation were evaluated in different cancers and were shown to have translational potential. In small cell lung cancer, Prexasertib was shown to enhance the response to PD-L1 blockade through STING-mediated T-cell activation[48]. Replication

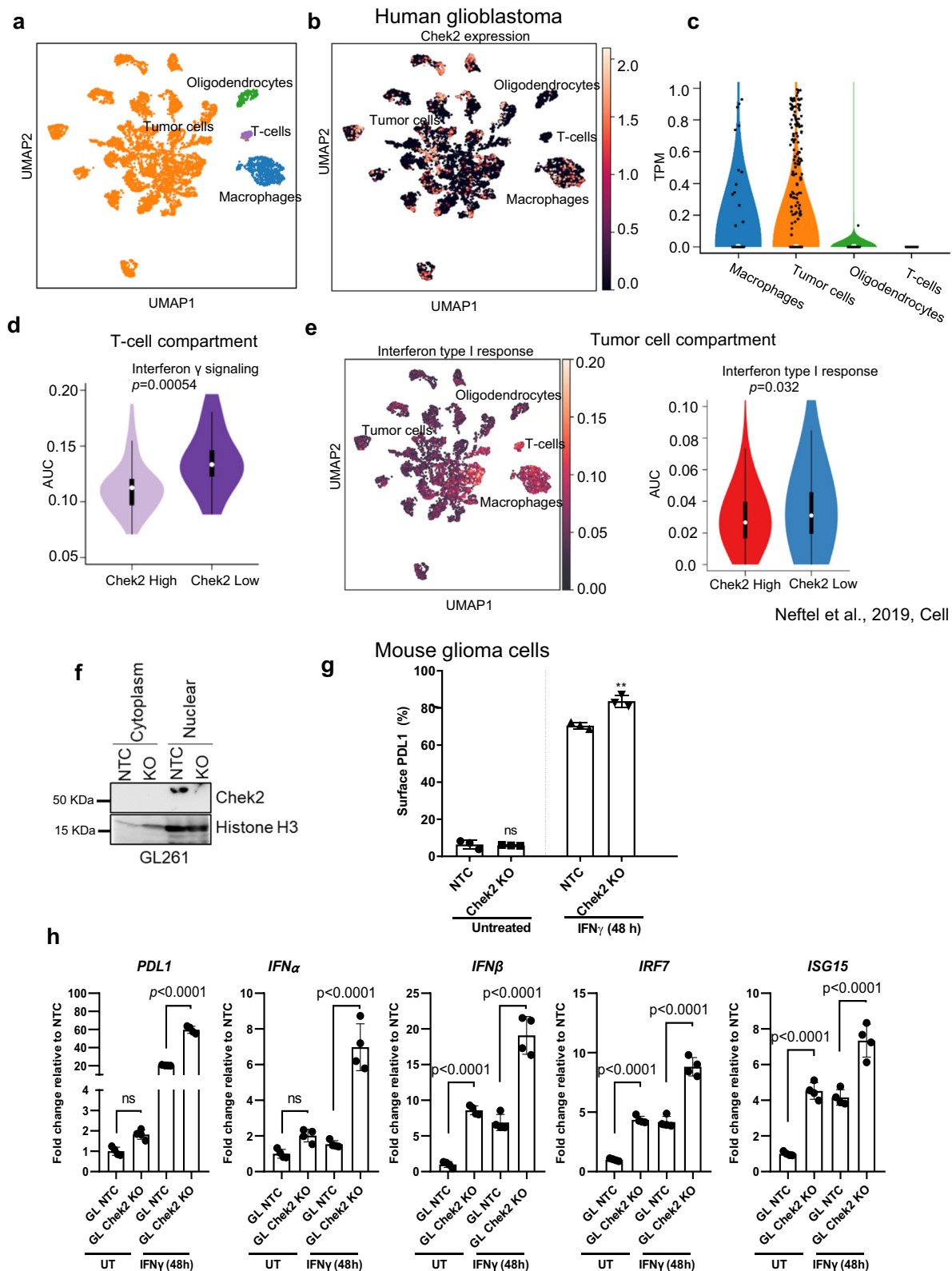

stress response defect genes with the accumulation of immunosti-mulatory cytosolic DNA are more likely to identify subjects who might benefit[49]. Pharmacological induction of replication stress responses with AZD7762 was shown to further expand the benefits of immune checkpoint blockade to more patients with other types of malignancies[49]. Also the Chek1/2 inhibitors used in this study target both Chek1 and Chek2. Whereas we cannot rule out the contribution of

inhibition of Chek1 when using these drugs, all our experimental evidence based on CRISPR screen, CD8 T- cell proliferation assay, survival study and scRNA-seq analysis confirms the role of Chek2 inhibition/depletion in modulating CD8 T-cell phenotype and response to immune checkpoint blockade, whereas the results for *Chek1* on the CRISPR screen and scRNA-seq analyses, did not suggest that this kinase is immune modulatory.

**Fig. 3 | *CHEK2* expression in tumor cells is inversely associated with type 1 interferon signaling in human and mouse gliomas. a** UMAP graph of all single cells, showing cell annotation for macrophages (blue), tumor cells (orange), oligodendrocytes (green) and T cells (purple). Each dot represents an individual cell, *n* = 7930 cells, from 28 human glioblastoma tumor samples. **b** UMAP plot showing the expression of *CHEK2* in macrophages, tumor cells, oligodendrocytes, and T cells. **c** Violin plot comparing the expression of *CHEK2* in macrophages, tumor cells, oligodendrocytes and T cells. **d** Violin plot (right) of the gene signature scores of Interferon γ signaling in *n* = 94 T cells from high *CHEK2* vs low *CHEK2* expressing cases. **e** UMAP (left) and violin plot (right) of the gene signature scores of Interferon type I response in *n* = 6,863 tumor cells with high and low expression of *CHEK2*. Median of the frequency of *CHEK2* expression (CPM > 0) inside the tumor cell compartment was used to dichotomize the 28 samples. The scRNA-seq data was used from the study published by Neftel et al.[28]. For (**d** and **e**), the *p* value represents two-tailed Mann–Whitney test; whiskers represent minimum and maximum values, the white dot inside the box represents the median and the box extends from the 25th to 75th percentiles. **f** Representative western blot showing knockout of Chek2 in GL261 cells. Histone H3 is used as a loading control. *N* = 3 independent replicates. **g** Flow cytometry analysis showing surface expression of PD-L1 on GL261 Chek2 KO and non-targeting control (NTC) clones, at the basal level and upon stimulation with IFNγ for 48 h. The histogram shows mean ± SE of one representative experiment of 3 independent experiments. *p* = 0.0034 by two-sided *t* test. **h** Quantitative real-time PCR analysis showing mean mRNA expression of PD-L1, IFN-α, IFN-β, IRF7 and ISG15 at the basal level and upon stimulation with IFN-γ for 48 h. GAPDH mRNA expression was used to normalize the expression of the target gene. Histograms represent mean ± SD, *p* value corresponds to *p* adjusted. *N* = 2 independent experiments with 4 technical replicates/independent experiment. Differences among cell types were evaluated using one-way ANOVA with post hoc Tukey's multiple comparisons test. GL NTC in the figure corresponds to GL261 NTC and GL Chek2 KO corresponds to GL261 Chek2 KO. Source data for **f**–**h** are provided as a Source Data file.

Our study raises the question of whether Chek2 modulation might enhance other forms of immunotherapies. Given that the discovery of Chek2 was a result of an unbiased CD8 T-cell screen, it would be worthwhile to test the therapeutic effect of Chek1/2 inhibition therapy in combination with other types of T-cell-based immunotherapies like Chimeric Antigen Receptor (CAR) T Cell, bispecific t cell engagers (BiTEs), B-cell vaccine (Bvax) immunotherapies in addition to CTLA-4 and PDL1 blockade therapies[50,51]. The combination of Chek2 inhibition therapy with oncolytic viral therapy can also be a rational approach, based on the premise that Chek2 depleted tumor cells perhaps resembled virus-infected cells, in the context of induction of type I interferon response. Hence it would be worthwhile to test if Chek2 inhibition improves the efficacy of oncolytic viral therapies, perhaps by amplifying the type I interferon signaling[52]. Along with Chek2, the CRISPR screen identified other kinases like Map3k7, TK1, Uck1, Nek6 and Nuak1 that might contribute to resistance to CD8 T-cell responses. These might warrant further investigation as potential targets to potentiate T-cell-based immunotherapies in GBM. Sensitivity conferring kinases like Rbks, Scyl1, Tbk1, Axl, Fgr, Prkch identified in this study can be further investigated in the context of predictive biomarkers for T-cell-based immunotherapies.

Our study has several limitations. Importantly, the causal, mechanistic experiments were performed using murine glioma models, and investigation of these mechanisms in human tumors is warranted. Indeed, while mouse models are useful for initial testing of novel therapeutics, they may fail to fully recapitulate the biology of human GBMs and hence response to the therapy. In particular, a recent analysis of 3 GBM tumors revealed absence of STING expression, in the context of promoter methylation of this gene[53]. Given that publicly available GBM scRNA-seq datasets show heterogeneous expression of STING in tumor cells across patients[28,29], how STING promoter methylation relates to activation of STING pathway when *CHEK2* is lost, remains to be determined.

In summary, this work proposes a rationale and preclinical evidence to combine Checkpoint 1/2 inhibition with PD-1/PD-L1 blockade, to enhance the responsiveness of gliomas to immune checkpoint blockade therapy.

## Methods
### Mouse models
All mice were housed at the Center for Comparative Medicine at Northwestern Feinberg School of Medicine. Mice were housed in a conventional barrier facility with 12-h light/12-h dark cycles and ad libitum access to food and water. C57BL/6 and CD8 KO mice were obtained from The Jackson Laboratory. All experiments were performed on 6–8 weeks old mice, age- and gender-matched. For all survival studies, C57BL/6 and CD8 KO mice were between 6–8 weeks old, and numbers of male/female mice were always equivalent

between control and experimental groups. All mouse protocols performed in this study were approved by Northwestern's Institutional Animal Care and Use Committee (IACUC) under study approval number IS00015286.

### Cell lines and tumor implantation
The HEK293T cells were purchased from ATCC (Cat no. CRL-3216). The GL261 mouse glioma cell line was purchased from the National Cancer Institute (NCI). Cells were cultured in Dulbecco's modified Eagle's medium (Corning) supplemented with 10% fetal bovine serum (FBS; HyClone) and penicillin-streptomycin. For ovalbumin overexpression, the pAc-Neo-OVA plasmid was purchased from Addgene[54]. GL261 NTC (non-targeting control) and GL261 Chek2 KO were transfected with the plasmid and were selected and maintained in G-418 (200 μg/ml; Sigma-Aldrich). NPA cells are highly malignant stem cells derived from NPA (Nras, shp53, shATRx, and IDH-Wild Type) gliomas and are a generous gift from Prof. Maria Castro. These cells grow as neurospheres in culture media (DMEM/F12 with 1× B27 supplement, 1× N2 supplement, 1× Normocin, and 1× antibiotic/antimycotic supplemented with human recombinant EGF and basic-FGF at concentrations of 20 ng/mL each). The HEK293T, GL261 and NPA cell lines used in this study were profiled for short tandem repeat (annually) and mycoplasma contamination (semiannually). All the assays were performed on STR tested and mycoplasma negative HEK293T, GL261 and NPA cell lines.

### Intracranial immunocompetent mouse model
The protocol (IS00015286) for intracranial implantation and monitoring of animals was approved by the IACUC at Northwestern University. Both wild-type C57BL/6 and CD8 KO Mice were housed in pathogen-free conditions at a relatively constant temperature of 24 °C and humidity of 30–50%. 6 to 8 weeks old male and female wild-type C57BL/6 mice purchased from the Charles River Laboratories and CD8 KO mice (B6.129S2-Cd8atm1Mak/J; Strain #:002665) purchased from the Jackson laboratory, were used in these studies. The protocol followed to generate intracranial models was as follows. The mice were anesthetized through intraperitoneal administration of a stock solution containing ketamine (100 mg/kg) and xylazine (10 mg/kg). The surgical site was disinfected and an incision was made at the midline for access to the skull[55]. A total of $5 × 10^4$ GL261 cells or NPA cells were implanted to develop orthotropic tumors for the listed cell lines. Typically for every intracranial implantation, 2.5 μl cell suspensions are prepared in sterile PBS and loaded into a 29 G Hamilton Syringe. Implantation was done slowly for three minutes into the left hemisphere of the mouse brain at 3 mm depth through a transcranial burr hole created 3 mm lateral and 2 mm caudal to the bregma. Following implantation, the incision was closed using 9 mm stainless steel wound clips, and each mouse was placed into a clean cage over a

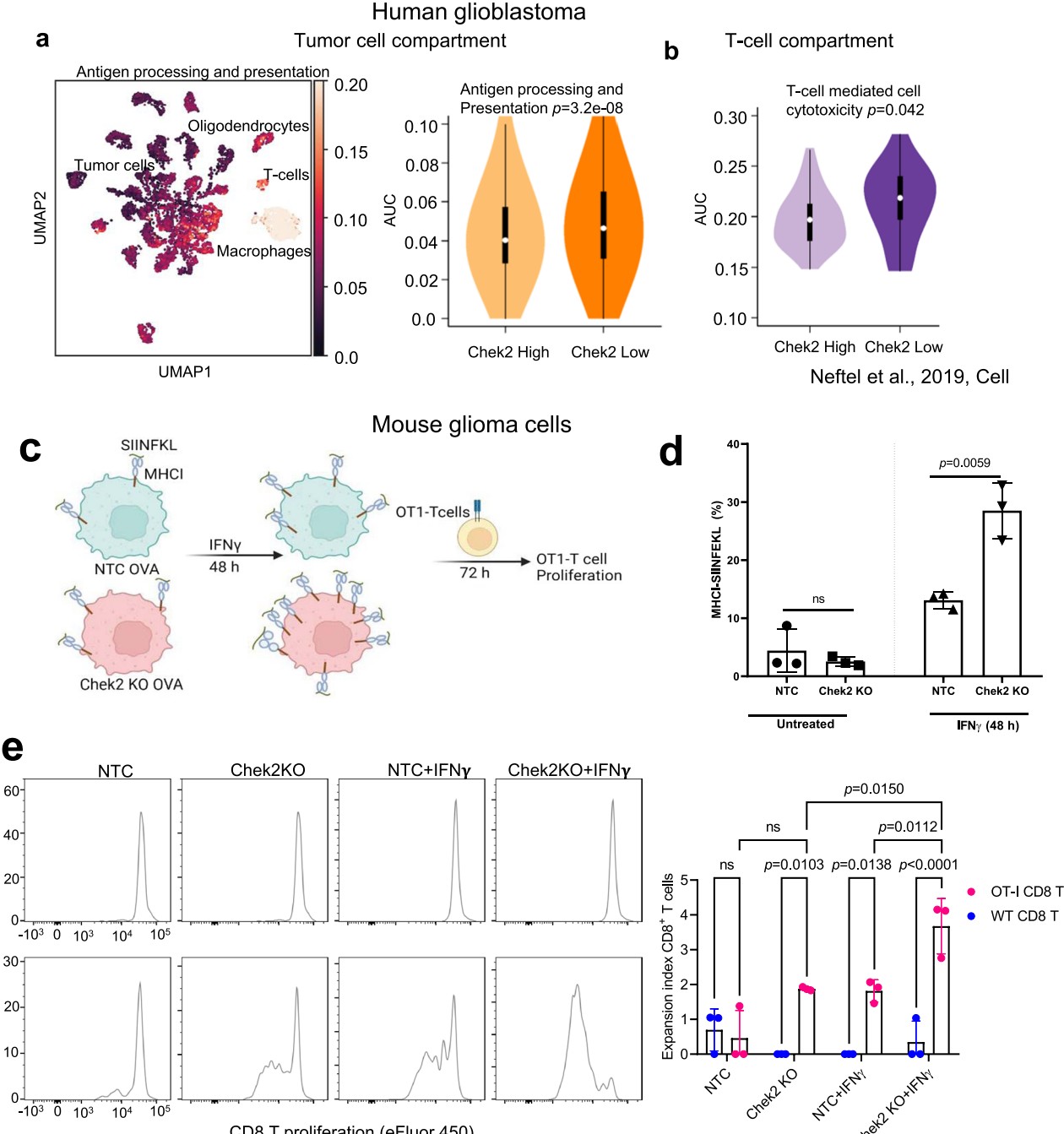

**Fig. 4 | *CHEK2* expression in tumor cells is inversely associated with enhanced antigen presentation on tumor cells, in human and mouse gliomas. a** UMAP (left) and violin plot (right) of the gene signature scores of antigen processing and presentation pathway in $n$ = 6,863 tumor cells high and low *CHEK2* expression from 28 human glioblastoma tumor samples. **b** Violin plot of the gene signature scores of T-cell-mediated cytotoxicity pathway in $n$ = 94 T cells from high *CHEK2* vs low *CHEK2* expressing samples. The scRNA-seq data was used from the study published by Neftel et al.[28]. For (**a** and **b**), the $p$ value represents two-tailed Mann–Whitney test; whiskers represent minimum and maximum values, the white dot inside the box represents the median and the box extends from the 25th to 75th percentiles. **c** Schematic showing the assay design to test the ability of Chek2 KO cells to promote OT-I CD8+ T-cell activation, assessed by cell proliferation (expansion index). **d** Flow cytometry analysis showing surface expression of MHCI-SIINFEKL on GL261 Chek2 KO and non-targeting control (NTC) clones, at the basal level and upon stimulation with IFNγ for 48 h. The histogram shows mean ± SD of one representative experiment of 3 independent experiments. $p$ = 0.0059 by two-sided $t$ test. **e** Four samples-NTC, Chek2 KO, NTC + IFNγ and Chek2 KO + IFNγ were cultured in individual well of the 6-well plates in triplicates. Each of the replicates of these treatment groups were cocultured with CD8+ T cells and OT-I CD8+ T cells independently ($N$ = 3/experimental condition). WT and OT-1 CD8+ T-cell proliferation in all 24 samples was assessed by eFluor 450 fluorescence dilution individually. For NTC + IFNγ and Chek2 KO + IFNγ groups, the respective clones were stimulated with IFNγ for 48 h prior to culturing with WT CD8+ T cells or OT-I CD8+ T cells. Histograms represent mean ± SD of $N$ = 3 replicates/condition. $p$ values by two-way ANOVA. Source data for **d**, **e** are provided as a Source Data file.

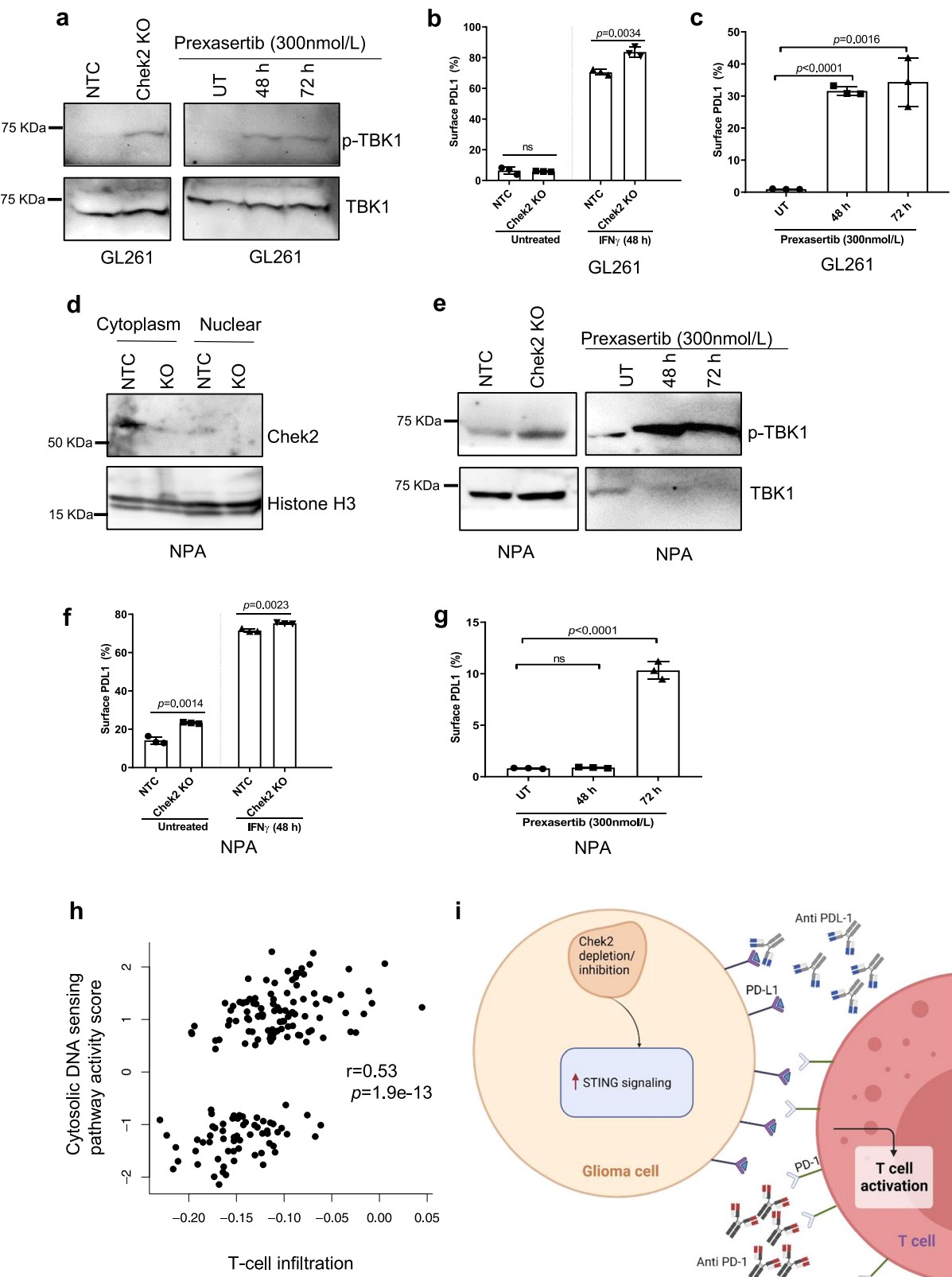

heating pad until recovery from anesthesia. Mice were monitored daily and were euthanized when they approached the endpoint (weight loss is >20% of pre-treatment body weight or loss of mobility or severe neurological disabilities such as seizure, circular motion, etc.) as described in the IACUC protocol. The male: female sex ratio of 1:1 was kept at the time of treatment randomization for all the experiments. The tumor injection site and depth were kept constant for all the intracranial experiments. Mice were monitored over the period of study and were euthanized when they approached the endpoint as described in the IACUC protocol (like loss of weight/mobility/body condition and severe neurological disabilities such as seizure, circular motion etc.).

**Fig. 5 | Chek2 depletion/inhibition leads to STING pathway activation in mouse glioma cells. a** Western blot showing the phosphorylation of TBK1 in GL261 Chek2 KO and non-targeting control (NTC) clones and in GL261 glioma cells treated with Chek1/Chek2 inhibitor Prexasertib (300 nmol/L) at the indicated time points. Total TBK1/β-Actin is used as a loading control. $N = 3$ independent replicates. Flow cytometry analysis showing surface expression of **b,** PD-L1 on GL261 Chek2 KO and NTC clones, at the basal level and upon stimulation with IFNγ for 48 h and **c**, PD-L1 on GL261 glioma cells treated with Prexasertib (300 nmol/L) at the indicated time points. **d** Western blot showing knockout of Chek2 in NPA cells. Histone H3 is used as a loading control. Western blot showing the phosphorylation of TBK1 in **e**, NPA Chek2 KO and NTC clones and in NPA glioma cells treated with Chek1/Chek2 inhibitor Prexasertib (300 nmol/L) at the indicated time points. Total TBK1 is used as a loading control. For **d**–**e**, $N = 3$ independent replicates. Flow cytometry analysis showing surface expression of **f**, PD-L1 on NPA Chek2 KO and NTC clones, at the basal level and upon stimulation with IFNγ for 48 h and **g**, PD-L1 on NPA glioma cells treated with Prexasertib (300 nmol/L) at the indicated time points. **h** Scatter plot showing correlation between T-cell infiltration and cytosolic DNA sensing-STING pathway in TCGA GBM dataset ($n = 167$ samples, two-tailed, Pearson correlation coefficient r = 0.53, $p = 1.9e–13$). **i** Proposed model: Chek2 depletion or pharmacological inhibition results in STING pathway activation and PD-L1 upregulation, which may sensitize gliomas to checkpoint blockade therapy. For **b**, **c**, **f** and **g** 10,000 cells/condition were analyzed over 3 independent experiments (mean ± SD, unpaired two-tailed $t$-test). Source data for Fig. **a**–**g** are provided as a Source Data file.

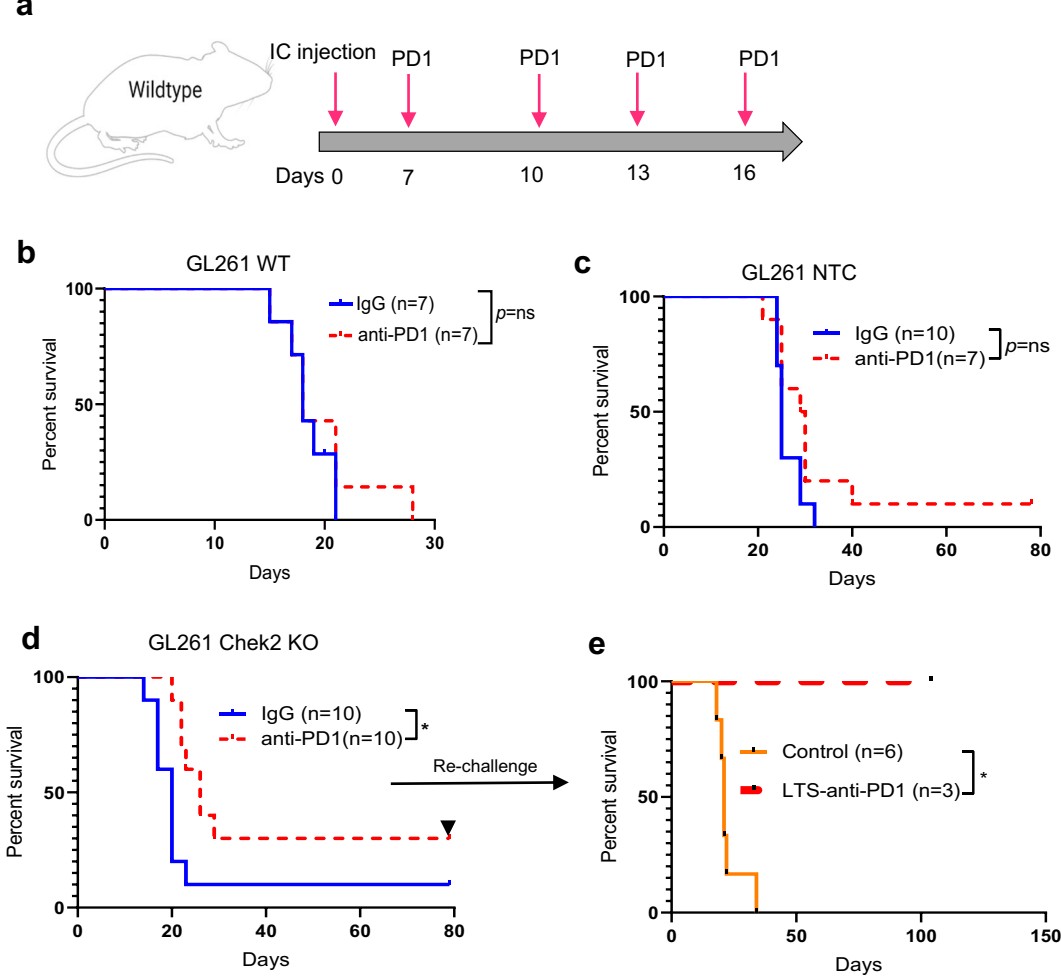

**Fig. 6 | Chek2 depleted gliomas show improved response to PD-1 blockade.**
**a** Dosing scheme for the PD-1 blockade treatment in mice with intracranially implanted GL261 glioma cells. **b** Kaplan–Meier (KM) survival curves of the C57BL/6 mice bearing GL261 wild-type cells. One group of mice ($n = 7$/group) were treated with the anti-PD-1 antibody and the other group of mice was treated with the isotype control antibody. 200,000 GL261 cells were injected/mouse. The median survival: IgG, 18 days; anti-PD-1, 18 days; Statistics: anti-PD-1 versus IgG, $p = 0.59$. **c** KM survival curves of the C57BL/6 mice bearing GL261 NTC (non-targeting control) cells. One group of mice ($n = 7$/group) was treated with the anti-PD-1 antibody, and the other group of mice ($n = 10$/group) was treated with the isotype control antibody. 50,000 GL261 NTC cells were injected/mouse. The median survival: IgG, 25 days; anti-PD-1, 29.5 days; Statistics: anti-PD-1 versus IgG, $p = 0.07$. **d** KM survival curves of the C57BL/6 mice bearing GL261 Chek2 KO cells. One group of mice ($n = 10$/group) was treated with the anti-PD-1 antibody, and the other group of mice was treated with isotype control antibody. 50,000 GL261 Chek2 KO cells were injected/mouse. The median survival: IgG, 20 days; anti-PD-1, 26 days; Statistics: anti-PD-1 versus IgG, $p = 0.01$. **e** The LTS in the Chek2 KO implanted group were rechallenged 80 days after the first implantation of Chek2 KO cells in the contralateral hemisphere with same type of cells and monitored. KM survival curves of the LTS mice from isotype and anti-PD-1 groups along with the control mice group which were implanted for the first time. The median survival: naive controls (6 mice), 21 days; isotype (1 mouse), undefined; anti-PD-1 (3 mice), undefined. Statistics: naive control versus isotype, $p = 0.11$; naive control versus anti-PD-1, $p = 0.01$. On the figure LTS is long-term survivors and anti-PD-1 is PD-1 blockade. Survival analysis for **b**–**e** was performed using the log-rank test. Statistical significance on the figure is depicted as ns: not statistically significant, $*p < 0.05$. Source data for **b**–**e** are provided as a Source Data file.

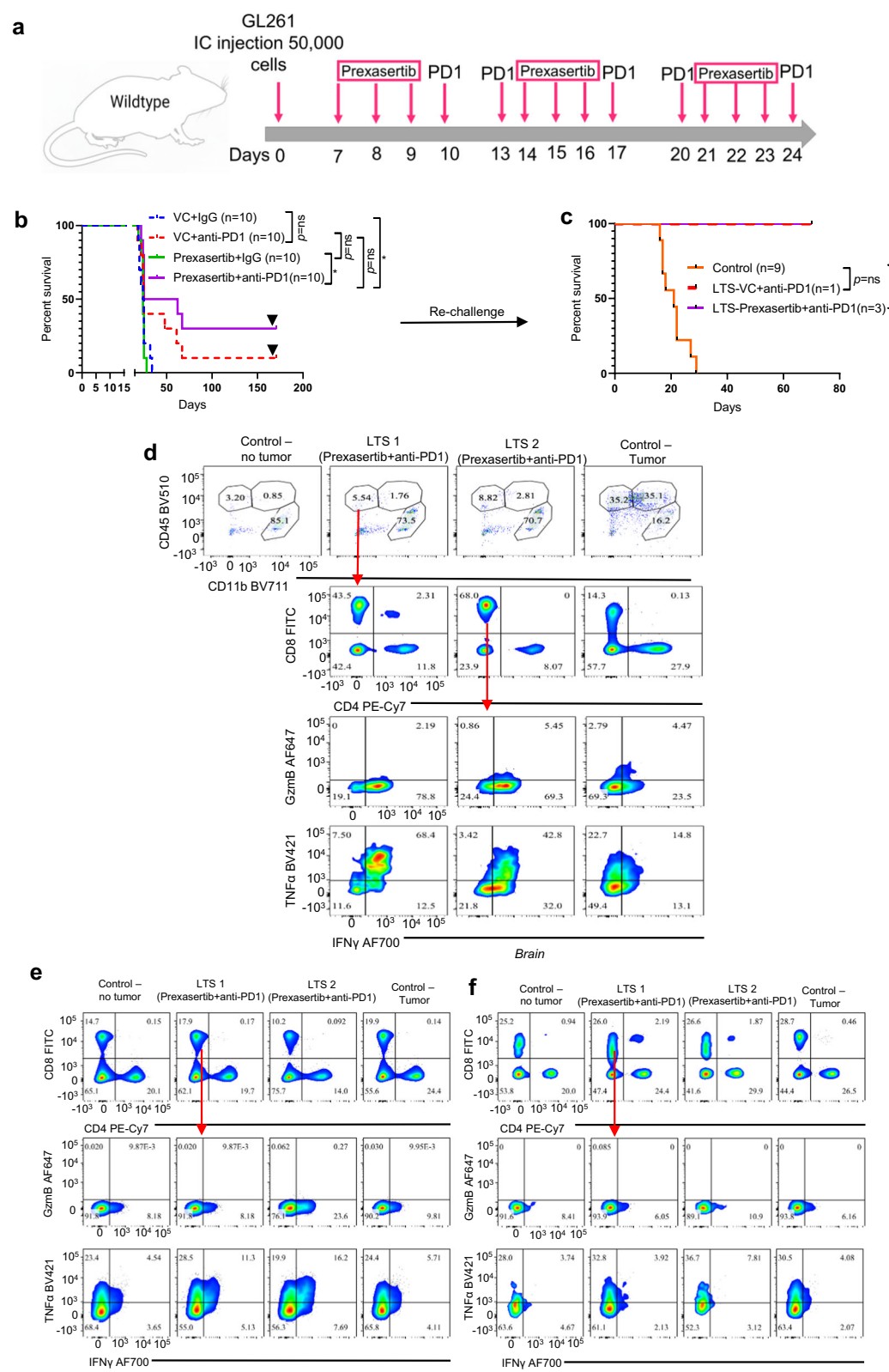

## In vivo kinome KO CRISPR screen

To perform the kinome KO CRISPR screen 1 and 2, the Brie kinome KO Library was purchased from Addgene[56]. The Brie kinome KO library included 714 kinases represented by 2856 guides and an additional 100 non-targeting controls. The library preparation, virus production, and multiplicity of infection (MOI) determination were done as described in[57] for the 2 CRISPR screens. Briefly, for the CRISPR screen 1, the virus was produced in HEK293T cells and the MOI was determined in 96-well plates using Promega® CellTiter-Glo®. Further, the GL261 cell line was transduced with the Brie kinome KO library at MOI < 0.3. The transduced cells were selected with 2 µg/ml puromycin for 7 days. The kinome KO GL261 cells (200,000 cells/mouse) were implanted intracranially in C57B/L6 mice with either fully competent immunity (n = 11) or in the CD8 KO (n = 11) background group at 733X representation of

**Fig. 7 | Combination of Prexasertib and PD-1 blockade improves survival in glioma-bearing mice. a** The schematic representation of the dosing scheme for the survival experiment. **b** KM survival curves for C57BL/6 mice bearing GL261 glioma. Seven days after intracranial tumor implantation, the animals were randomized into 4 groups (10 animals/group): vehicle and isotype control (IgG), anti-PD-1, Prexasertib (Chek1/2 inhibitor), and Prexasertib and the anti-PD-1 combination group. The median survival duration in the treatment groups were as follows: VC + IgG, 24.5 days; VC + anti-PD-1, 25 days; Prexasertib + IgG, 24 days; Prexasertib + anti-PD-1, 43.5 days. Statistics: VC + IgG vs VC + anti-PD-1, $p = 0.09$; VC + IgG vs Prexasertib + IgG, $p = 0.5$; and VC + IgG vs Prexasertib + anti-PD-1, $p = 0.01$. **c,** KM survival curves for long-term survivors and naive controls that were rechallenged in the contralateral hemisphere. The median survival in the treatment groups were as follows: naive controls (9 mice), 21 days; VC + anti-PD-1 (1 mouse), undefined; Prexasertib + anti-PD-1 (3 mice), undefined. Statistics: naive control vs VC + anti-PD-1,

$p = 0.09$; naive control vs Prexasertib+ anti-PD-1, $p = 0.0062$. Survival analysis for **b**, **c** was performed using the log-rank test. **d** Analysis of CD8 T-cell phenotype in the long-term survivors (LTS). Freshly dissected brains from Prexasertib + anti-PD-1 group, naive control (no tumor), and glioma-bearing control mice were analyzed for CD8 T-cell phenotype using flow cytometry. **e** Analysis of CD8 T-cell phenotype in the long-term survivors. Freshly dissected spleens from Prexasertib + anti-PD-1 group, naive control (no tumor), and glioma-bearing control mice were analyzed for CD8 T-cell phenotype using flow cytometry. **f** Analysis of CD8 T-cell phenotype in the long-term survivors. Freshly dissected deep cervical lymph nodes (dCLN) from Prexasertib + anti-PD-1 group, naive control (no tumor), and glioma-bearing control mice were analyzed for CD8 T-cell phenotype using flow cytometry. Statistical significance on the figure is depicted as ns: not statistically significant, *$p < 0.05$. Source data for **b**, **c** are provided as a Source Data file.

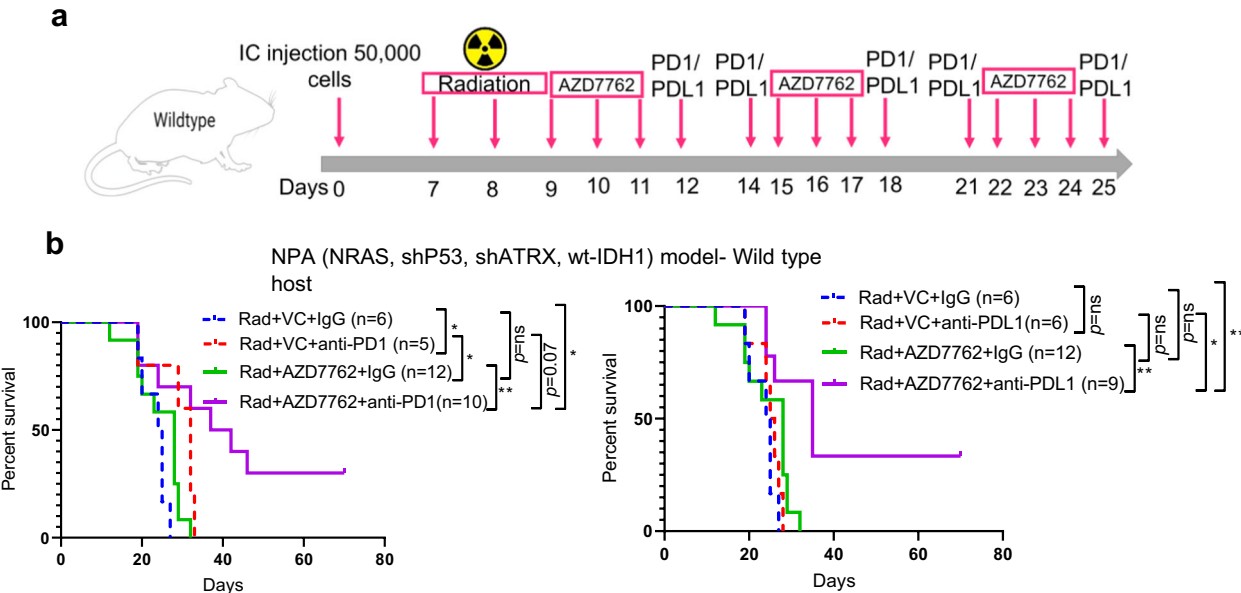

**Fig. 8 | The combination of radiotherapy, AZD7762 and PD-1/PD-L1 blockade improves the survival of glioma-bearing mice in NPA glioma model. a** The schematic representation of the dosing scheme followed for the survival experiment. **b** (Left) KM survival curves for NPA glioma-bearing C57BL/6 mice. Seven days after intracranial implantation, all the animals were irradiated with 3 Gy whole-brain radiation for 3 consecutive days. 9 days after tumor implantation, the animals were randomized into 4 groups: vehicle control ($n = 6$), anti-PD-1 ($n = 5$), AZD7762 (Chek1/2 inhibitor) ($n = 12$) and AZD7762 + anti-PD-1 ($n = 10$) combination group. Survival analysis was performed using the log-rank test. The median survival duration in the treatment groups were as follows: Rad + VC + IgG, 24.5 days; Rad + VC + anti-PD-1, 32 days; Rad + AZD7762 + IgG, 28 days; Rad + AZD7762 + anti-PD-1, 39.5 days. Statistics: Rad + VC + IgG vs Rad + VC + anti-PD-1, $p = 0.02$; Rad + VC + IgG vs Rad + AZD7762 + IgG, $p = 0.09$; and Rad + VC + IgG versus Rad + AZD7762 + anti-

PD-1, $p = 0.018$. (Right) KM survival curves for NPA glioma-bearing C57BL/6 mice. 7 days after intracranial implantation, all the animals were irradiated with 3 Gy whole-brain radiation for 3 consecutive days. 9 days after tumor implantation, the animals were randomized into 4 groups: vehicle control ($n = 6$), anti-PD-L1 ($n = 6$), AZD7762 (Chek1/2 inhibitor) ($n = 12$) and AZD7762 + anti-PD-L1 ($n = 9$) combination group. Survival analysis was performed using the log-rank test. The median survival duration in the treatment groups were as follows: Rad + VC + IgG, 24.5 days; Rad + VC + anti-PD-L1, 25.5 days; Rad + AZD7762 + IgG, 28 days; Rad + AZD7762 + anti-PD-L1, 35 days. Statistics: Rad + VC + IgG vs Rad + VC + anti-PD-L1, $p = 0.32$; Rad + VC + IgG vs Rad + AZD7762 + IgG, $p = 0.09$; and Rad + VC + IgG versus Rad + AZD7762 + anti-PD-L1, $p = 0.0065$. Statistical significance on the figure is depicted as ns: not statistically significant, *$p < 0.05$, **$p < 0.01$. Source data for **b** are provided as a Source Data file.

the kinome KO library/group, that is 2.2 million cells distributed across 11 animals. The number of expected guides/mouse is 200,000 guides. 2/11 animals from the CD8 KO group did not recover post intracranial injection, hence CD8 KO group had $n = 9$ animals, which corresponds to a 600X representation of the kinome KO library/group. For the CRISPR screen 2, GL261 mouse glioma cells were transduced with kinome knockout library, and the transformed cells were implanted in WT ($n = 11$) and CD8 KO ($n = 12$) mice. The library representation of 733X was maintained in the WT and 800X in the CD8 KO group by injecting 200,000 cells/mouse. The WT animal group received Isotype antibody (200 μg /i.p./dose of InVivoMAb rat IgG2a isotype control, BXCELL) twice a week for two weeks and the CD8 KO group did not receive any treatment. As the animals approached the endpoint, they were sacrificed, and the genomic DNA was extracted from the tumor

region. The animals sacrificed at the early stage between days 18 and 23 were uniquely barcoded as well as animals sacrificed between days 24 and 38 from both WT and CD8 KO hosts were uniquely barcoded, guides were amplified and sequenced. The animals on the CRISPR screen were sacrificed as they approached their endpoint and the gDNA from the glioma region was extracted with the Zymo Research Quick-DNA midiprep plus kit. The sgRNA was amplified with the unique barcode primer (Supplementary Table 1). The sgRNAs were pooled together and sequenced in a Next-generation sequencer (Next Seq). The samples were sequenced according to the Illumina user manual with 80 cycles of read 1 (forward) and 8 cycles of index 1. 20% PhiX was added in the Next seq to improve library diversity and aim for coverage of >1000 reads per sgRNA. The demultiplexing and alignments of the sequenced sgRNAs were done by the Quantitative Data

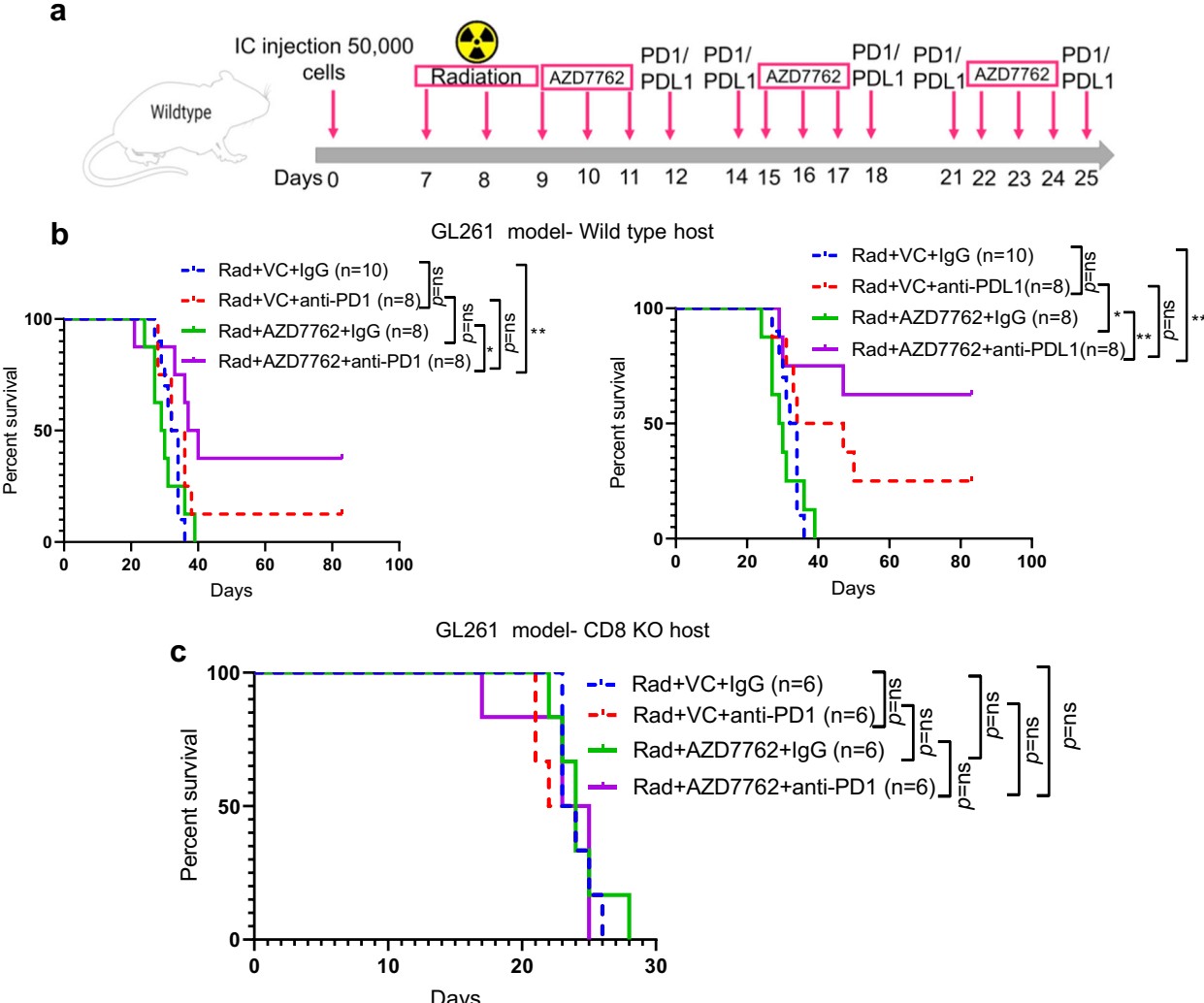

**Fig. 9 | The combination of radiotherapy, AZD7762 and PD-1/PD-L1 blockade improves the survival of glioma-bearing mice in GL261 glioma model. a** The schematic representation of the dosing scheme followed for the survival experiment. **b** (Left) KM survival curves for GL261 glioma-bearing C57BL/6 mice. Survival analysis was performed using the log-rank test. The median survival duration in the treatment groups were as follows: Rad + VC + IgG, 33 days; Rad + VC + anti-PD-1, 34 days; Rad + AZD7762 + IgG, 29.5 days; Rad + AZD7762 + anti-PD-1, 38.5 days. Statistics: Rad + VC + IgG vs Rad + VC + anti-PD-1, $p = 0.16$; Rad + VC + IgG vs Rad + AZD7762 + IgG, $p = 0.71$; and Rad + VC + IgG vs Rad + AZD7762 + anti-PD-1, $p = 0.008$. (Right) KM survival curves for GL261 glioma-bearing C57BL/6 mice. Survival analysis was performed using the log-rank test. The median survival duration in the treatment groups were as follows: Rad + VC + IgG, 33 days; Rad +

VC + anti-PD-L1, 40.5 days; Rad + AZD7762 + IgG, 29.5 days; Rad + AZD7762 + anti-PD-L1, undefined. Statistics: Rad + VC + IgG vs Rad + VC + anti-PD-L1, $p = 0.059$; Rad + VC + IgG vs Rad + AZD7762 + IgG, $p = 0.71$; and Rad + VC + IgG vs Rad + AZD7762 + anti-PD-L1, $p = 0.0062$. **c** KM survival curves for GL261 glioma-bearing CD8 KO mice. Survival analysis was performed using the log-rank test. The median survival duration in the treatment groups were as follows: Rad + VC + IgG, 23.5 days; Rad + VC + anti-PD-1, 23 days; Rad + AZD7762 + IgG, 24 days; Rad + AZD7762 + anti-PD-1, 24 days. Statistics: Rad + VC + IgG vs Rad + VC + anti-PD-1, $p = 0.66$; Rad + VC + IgG vs Rad + AZD7762 + IgG, $p = 0.7$; and Rad + VC + IgG vs Rad + AZD7762 + anti-PD-1, $p = 0.78$. Statistical significance on the figure is depicted as ns: not statistically significant, *$p < 0.05$, **$p < 0.01$. Source data for **b**, **c** are provided as a Source Data file.

Science Core facility (Northwestern University). Guides with raw read counts <40 were excluded from the analysis. Normalized read counts were obtained by normalizing to total read count per sample (normalized reads per sgRNA = reads per sgRNA/total reads for all sgRNAs in the sample × $10^6$ + 1)[58]. The fold change, which is higher than the most enriched non-targeting sgRNA, and fold change, which is lower than the most depleted non-targeting sgRNA, were marked as enriched and depleted respectively in the analysis.

### Generation of knockout and overexpression clones

Single gene knockout clones were generated in lentiCRISPRv2 (one vector system). The vector backbone was purchased from Addgene (lentiCRISPRv2 was a gift from Feng Zhang (Addgene plasmid # 52961; http://n2t.net/addgene:52961; RRID:Addgene_52961))[59] and

the protocol for guide cloning and generation of the virus was as described in[59]. The Chek2 KO and control clones were selected using puromycin from sigma, GL261 (2 μg/ml puromycin) and NPA (1 μg/ml puromycin). The Chek2 KO was confirmed using western blotting (Chek2 antibody cell signaling technology, 1:200 dilution Cat no. 2662). The guide sequences for Chek2 KO and NTC are shown in Supplementary Table 2A. For OVA overexpression, the pAc-Neo-OVA plasmid was purchased from Addgene[54]. GL261 NTC and GL261 Chek2 KO cells were transfected with the pAc-Neo-OVA plasmid using Lipofectamine 2000 reagent (Fisher Scientific Cat no.11668027). The transfected cells were selected for 10 days in 200 μg/ml of G-418 (Sigma-Aldrich Cat no. G8168-10ML). The OVA expressing cells were always maintained in 200 μg/ml of G-418 selection pressure for all the assays.

## Quantitative real-time PCR

The GL261 NTC and Chek2 KO cells were treated with IFN-γ for 48 h and then the RNA was extracted, cDNA was made, and real-time PCR was performed using SYBR green. The primer sequences for the mouse cytokine signatures were adopted from ref. [60]. The table for primer sequences is listed in Supplementary Table 2B.

## Murine immunophenotypic analysis

For immunophenotyping assay, tumor-bearing mice were euthanized in a $CO_2$ chamber and intracardially perfused with chilled PBS. The brain was harvested and the single-cell suspensions were obtained by mechanical dissociation using a manual tissue homogenizer (Potter-Elvehjem PTFE pestle, Sigma-Aldrich) in HBSS. Myelin and debris were removed by Percoll gradient separation. Leukocytes from deep and superficial cervical lymph nodes were obtained by mechanical tissue dissociated using a 70-mm cell strainer and a syringe plunger. The leukocytes from brain, deep cervical lymph nodes (dCLN) and spleen were processed and stained for immunophenotyping[61]. Expression levels of granzyme B (GZMB), TNF-α, and IFN-γ by CD8 T cells in the brain of glioma-bearing mice were analyzed by flow cytometry. The CD16/32 (clone 93, 1:100 dilution) antibody was purchased from ebioscience. Mouse antibodies from BioLegend were used at 1:100 dilution: CD45 BV510 (clone 30F 11), CD11b BV711 (clone M1/70), CD8a FITC (Clone 5H10-1), IFN-γ AF700 (clone XMG1.2), GZMB AF647 (clone GB11), CD4 PE-Cy7 (clone GK1.5) and TNF-α BV421 (clone MP6-XT22). Dead cells and debris were labeled using the eBioscience Fixable viability dye eFluor780 (1:1000 dilution) (Thermo Fisher). For the in vitro assay, the cells were detached using EDTA to preserve the surface proteins and then stained for flow cytometry. The antibodies used were PD-L1 Pecy7 (Biolegend Clone 10F.9G2, 1:100 dilution) and MHCI-SIINFEKL APC (Biolegend Clone 25-D1.16, 1:100 dilution) for determination of the surface expression of these proteins upon stimulation with IFN-γ (100 μg/ml). For OT-1 cell proliferation assay, eBioscience Cell Proliferation Dye eFluor 450 (5 μM concentration) (Thermo Fisher) was used. Cells were acquired by the BD Symphony and analyzed by FlowJo™ v10.7.1 software.

## OT-1 cell proliferation assay

Mouse T cells were isolated from spleens of WT and OT-1 mice and enriched using the Mouse T-cell isolation kit (StemCell Technologies, Cat no. 19853). The T cells were labeled with 10 μM of the eBioscience™ cell proliferation dye eFluor 450 (Thermo Fisher). Cells were activated with T-cell activator anti-CD3/CD28 beads (Dynabeads, Invitrogen, Thermo Fisher, Cat no. 11456D) at 1:3 beads:T-cell ratio supplemented with IL2 (50 U/mL; Peprotech, Cat no. 212-12)[51]. The WT and OT-1 T cells were labeled with eBioscience Cell Proliferation Dye eFluor 450 (5 μM concentration) (Thermo Fisher) and then cocultured at a 5:1 ratio (T cells: Tumor cells) with GL261 NTC and Chek2 KO clones with or without prior treatment with IFNγ for 48 h. The coculture was setup in a 96-well plate and kept for 72 h in the $CO_2$ incubator. Post 72 h all the T cells were collected, blocked with CD16/32 and stained with CD8a BV605 antibody (Biolegend Clone 53-6.7, 1:100 dilution) and Fixable viability dye eFluor780. The stained cells were acquired by the BD Symphony and analyzed by FlowJo™ v10.7.1 software.

## Therapy and dosing for mice survival studies

To test the effect of radiation (RT), chek1/2 inhibitors, and anti-PD-1 in animal survival, mice received total brain irradiation (Gammacell 40 Exactor, Best Theratonics) at 9 Gy in total for 3 days starting 7 days post tumor implantation. On the 9th day post tumor implantation, the animals were randomized into 4 groups: vehicle control group, anti-PD-1 group (200 μg /i.p./dose of InVivoMAb anti-mouse PD-1 (CD279), BXCELL), AZD7762 group (15 mg/kg/i.p./dose) and anti-PD-1 + AZD7762 combination group. The vehicle control group and the AZD7762 group received an isotype antibody (200 μg /i.p./dose of

InVivoMAb rat IgG2a isotype control, BXCELL) on days where the anti-PD-1 and combination groups received anti-mouse PD-1. The scheme for the administration of Chek1/2 inhibitor and PD-1 blockade is shown alongside the survival curves in the figures. For the survival study without radiation therapy, the animals were randomized into 4 groups: vehicle control, anti-PD-1 (200 μg/dose), Prexasertib (15 mg/kg/subcutaneous/dose), and anti-PD-1 + Prexasertib combination group. The treatment was started 7 days after tumor implantation. For all the rechallenge experiments, the long-term survivors and the control group did not receive any treatment.

## Drug preparation for in vivo dosing

AZD7762[36] and AZD1390 were dissolved in 0.9% saline containing 11.3% (2-hydroxypropyl)-β-cyclodextrin (Sigma Aldrich, H107-5G) on a magnetic stirrer for 30 min and stored at 4 °C to be used within 14 days. The vehicle control mice in the survival study received 11.3% (2-hydroxypropyl)-β-cyclodextrin. Prexasertib was prepared in 20% captisol (Cat no. RC-0C7-100) and the vehicle control mice in that survival study received 20% captisol[62].

## ScRNA-seq analysis

Single-cell RNA-seq data was obtained from 28 GBM patients previously published by Neftel et al.[28] GEO #GSE131928. All the quality control and filtration parameters as described in the original paper were retained. The markers used to define the cell types were SOX2 for tumor cells, CD14/CD68 for macrophages, CD3D for T cells, CD79A for B cells, MBP for oligodendrocytes, PECAM1 for endothelial cells and PDGFRB for pericytes. For visualization: the normalized gene barcode matrix was used to compute a neighborhood graph of cells, then Uniform Manifold Approximation and Projection (UMAP) was performed with default parameters. The whole pipeline was implemented using Scanpy[63]. As per the previous analysis by Neftel et al.[28] four cell types were identified as macrophages, tumor cells, oligodendrocytes and T cells. Differential expression in the tumor cell compartment was calculated by computing Welch's t-statistic between tumor cells with (CPM > 0) and without (CPM = 0) Chek2 expression. Median of the frequency of Chek2 expressing (CPM > 0) tumor cells inside the tumor cell compartment was used as a cutoff for dichotomizing high and low Chek2 expression in the 28 samples. Then T cells from each group were merged together for differential expression analysis. Ninety-four T cells were analyzed from the dataset[28]. Another independent scRNA-seq dataset from Abdel-fattah et al.[29] GEO series GSE182109 was used to validate the findings from Neftel et al.28 dataset. The same analysis parameters and cutoffs were used for both scRNA-seq datasets. 7767T cells were analyzed from the GSE182109 dataset[29]. R package AUCell 1.21.2 was used to calculate enrichment score for each of the gene sets in scRNA-seq data[64]. AUCell calculates "Area Under the Curve" (AUC) to assess the enrichment of the input gene set within the expressed genes for each cell using ranking based score method. Gene sets were obtained from MSigDB v7.1, utilizing the C5: GO gene sets collection[65]. The list of gene sets used in this study is available in the Supplementary Data 1.

## DNA whole exome and RNA sequencing for neoantigen prediction

The Chek2 KO and NTC cells were subjected to exome and RNA sequencing. Libraries were captured using the Agilent Mouse Exome reagent. Sequencing was performed on an Illumina HiSeq2000 (Illumina Inc.). The predictions for endogenous neoantigens were generated for GL261 glioma model using a cancer immunogenomics approach[66].

## Statistical analyses

Flow cytometry data was analyzed by FlowJo™ v10.7.1 software. Statistical analyses were performed using Graphpad Software (Prism v7.03). Student's t-test was used to measure statistical differences

between two groups. One-way or two-way analysis of variance was used for multiple comparisons, and $p$ values were adjusted for multiple comparisons where appropriate. Survival curves were generated via the Kaplan–Meier method and compared by the log-rank test. All the tests were two-sided and $p$ values less than 0.05 were considered significant. R package AUCell 1.21.2 was used to calculate enrichment score for each of the gene sets in scRNA-seq data.

## Data availability

The raw data related to CRISPR screens, RNA and Exome sequencing is available on the publicly available repository, Sequence Read Archive (SRA) as BioProject ID PRJNA822842. The Neftel et al.[28] scRNA-seq publicly available data used in this study is available through GEO GSE131928 and the Abdelfattah et al.[29] scRNA-seq publicly available data used in this study is available through GEO series GSE182109. The results shown in Fig. 5h are based upon the data generated by the TCGA Research Network (https://portal.gdc.cancer.gov/). All remaining data supporting the findings of this study are available within the article and its supplementary information files. Source data are provided with this paper.

## Code availability

The codes are available in https://github.com/RabadanLab/Chek2_scRNAseq.

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

## Acknowledgements

This work was funded by the NIH grants 1R01NS110703 (to AMS), 5DP5OD021356 (to A.M.S.), funding support from the Lou and Jean Malnati Brain Tumor Institute (to A.M.S.), and philanthropic support from Dan and Sharon Moceri (to A.M.S.), R01 CA120813 (to A.B.H.), P50CA221747 SPORE for Translational Approaches to Brain Cancer (to ML), DOD Horizon award W81XWH-20-1-0850 Log# CA190063 (to C.D.). We sincerely thank our deceased colleague Joshua Robert Kane for his contributions towards performing experiments and data analyses. The images in Figs. 1b, 2a, 4c, 5i, 6a–9a and Supplementary Figs. 11a, 12 and 13 were created with BioRender.com.

## Author contributions

C.D., C.L.C., and A. M. S., designed the experiments and wrote the manuscript. C.D., J.Z., L.C., A.G., B.C., D.Y.Z, D.K., K.K., R.Y., C.L.C performed the experiments and data analysis. C.D., J.Z., L.C., C.L., V.A.A., S.J.K, J.F., B.C., K.H., R.S., C.L.C, analyzed the data. M.S. and E.B. performed bioinformatic analysis on CRISPR screen data. P.Z. and J.M. performed radiotherapy on mice. P.H., M.S.A., G.D., R.R., M.L., A.B.H., and A.M.S. directed the research and provided the reagents. A.M.S. and C.D. generated the funding. All authors read and edited the manuscript.

## Competing interests

C.D., A.M.S., C.L-C, and L.C. are co-authors for the following patent filed by Northwestern University: Method of using checkpoint kinase 1/2 inhibitor therapy to modulate anti-tumoral response against cancer and sensitize gliomas to immunotherapy (US Patent App. 16/951,638). A.M.S. has received in-kind and or funding support for research from Agenus, BMS, and Carthera. R.R. is a member of the SAB of AimedBio, consultant for Arquimea Research and a founder of Genotwin. The remaining authors declare no other competing interests.

## Additional information

[1]Department of Neurological Surgery, Feinberg School of Medicine, Northwestern University, Chicago, IL, USA. [2]Northwestern Medicine Malnati Brain Tumor Institute of the Lurie Comprehensive Cancer Center, Feinberg School of Medicine, Northwestern University, Chicago, IL, USA. [3]Program for Mathematical Genomics, Department of Systems Biology, Columbia University, New York, NY, USA. [4]Department of Biomedical Informatics, Columbia University, New York, NY, USA. [5]Section of Neurological Surgery, University of Chicago Medicine, Chicago, IL, USA. [6]PECEM, Facultad de Medicina, Universidad Nacional Autónoma de México, Mexico City, Mexico. [7]Department of Neurological Surgery, Washington University School of Medicine, St Louis, MO, USA. [8]Department of Pathology and Immunology, Washington University School of Medicine, St Louis, MO, USA. [9]The Alvin J. Siteman Cancer Center at Barnes-Jewish Hospital and Washington University School of Medicine, St Louis, MO, USA. [10]Department of Biochemistry and Molecular Genetics, Northwestern University, Chicago, IL, USA. [11]NUSeq Core, Center for Genetic Medicine, Feinberg School of Medicine, Northwestern University, Chicago, IL, USA. [12]Innovative Genomics Institute, University of California, Berkeley, Berkeley, CA, USA. [13]Department of Bioengineering, University of California, Berkeley, Berkeley, CA, USA. [14]Center for Computational Biology, University of California, Berkeley, Berkeley, CA, USA. [15]Department of Neurosurgery, University of Michigan Medical School, Ann Arbor, MI, USA. [16]Department of Cell and Developmental Biology, University of Michigan Medical School, Ann Arbor, MI, USA. [17]Department of Neurology, Department of Pathology, Institute for Cancer Genetics, Columbia University Medical Center, New York, NY, USA. [18]These authors contributed equally: Crismita Dmello, Junfei Zhao. [19]These authors jointly supervised this work: Crismita Dmello, Catalina Lee-Chang, Adam M. Sonabend. ✉e-mail: crismita.dmello@northwestern.edu; catalina.leechang@northwestern.edu; adam.sonabend@northwestern.edu

