## [Peer Review File · Nature Communications]

Checkpoint kinase 1/2 inhibition potentiates anti-tumoral immune response and sensitizes gliomas to immune checkpoint blockadeEditorial Note: This manuscript has been previously reviewed at another journal that is not operating a transparent peer review scheme. This document only contains reviewer comments and rebuttal letters for versions considered at Nature Communications.

REVIEWERS' COMMENTS

Reviewer #1 (Remarks to the Author):

The updated manuscript from Dmello et al is significantly improved with the new experiments and analyses, as well as the textual revisions. The evidence presented is supportive of a tumor-intrinsic immunosuppressive function for Chk2, possibly due to suppression of PD-L1 and/or STING pathway activation. If the authors were able to address whether these effects are causative or correlative with tumor response, the mechanistic aspect of the study would be more robust. Does knockout of STING or PD-L1 result in loss of response to Chk2 inhibitor/anti-PD1 combination therapy? As a minor point regarding the description of Figure 8 studies, I would recommend stating that the comparison group is treated with standard of care radiotherapy with vehicle control. This is clear in the figure but as stated was not clear in the text (it seemed like the control group may not have received RT). Overall, the study is well conducted and has clear translational relevance. Thus, I am supportive of acceptance if these critiques can be addressed.

Reviewer #2 (Remarks to the Author):

The authors have addressed all concerns raised in the initial review. Inclusion of additional mouse models has strengthened the conclusions from the previous submission.

Reviewer #3 (Remarks to the Author):

The authors have adequately addressed all concerns raised in the initial review.

Second Revision

Reviewer #1 (Remarks to the Author)

The updated manuscript from Dmello et al is significantly improved with the new experiments and analyses, as well as the textual revisions. The evidence presented is supportive of a tumor-intrinsic immunosuppressive function for Chk2, possibly due to suppression of PD-L1 and/or STING pathway activation. If the authors were able to address whether these effects are causative or correlative with tumor response, the mechanistic aspect of the study would be more robust. Does knockout of STING or PD-L1 result in loss of response to Chk2 inhibitor/anti-PD1 combination therapy? As a minor point regarding the description of Figure 8 studies, I would recommend stating that the comparison group is treated with standard of care radiotherapy with vehicle control. This is clear in the figure but as stated was not clear in the text (it seemed like the control group may not have received RT). Overall, the study is well conducted and has clear translational relevance. Thus, I am supportive of acceptance if these critiques can be addressed.

Response: We thank the reviewer for reviewing our manuscript. In the revised manuscript we have shown that Chk2 depletion/inhibition results in increase activation of STING and upregulation of PD-L1 surface expression. Understanding the causative/correlative role of STING activation/PD-L1 over-expression as a mediator of tumor response downstream of Chk2 depletion/inhibition, is yet another claim that was not part of the initial manuscript. Although, we will be pursuing the investigation of the STING activation/PD-L1 over-expression in potentiating response to immune checkpoint blockade in presence of Chk2 inhibition, we consider this outside of the scope of our initial manuscript.

We thank the reviewer for the suggestion. In the revised manuscript, we have corrected the statement to indicate that the control group is treated with standard of care radiotherapy with vehicle control.

Reviewer #2 (Remarks to the Author)

The authors have addressed all concerns raised in the initial review. Inclusion of additional mouse models has strengthened the conclusions from the previous submission.

Response: We thank the reviewer for reviewing our manuscript.

Reviewer #3 (Remarks to the Author)

The authors have adequately addressed all concerns raised in the initial review.

Response: We thank the reviewer for reviewing our manuscript.